# Evaluation of Tumor Grade and Proliferation Indices before and after Short-Course Anti-Inflammatory Prednisone Therapy in Canine Cutaneous Mast Cell Tumors: A Pilot Study

**DOI:** 10.3390/vetsci9060277

**Published:** 2022-06-07

**Authors:** Shawna Klahn, Nikolaos Dervisis, Kevin Lahmers, Marian Benitez

**Affiliations:** 1Department of Small Animal Clinical Sciences, Virginia Maryland College of Veterinary Medicine, Virginia Tech, 245 Duck Pond Drive, Blacksburg, VA 24061, USA; dervisis@vt.edu (N.D.); marian.benitez19@gmail.com (M.B.); 2Department of Biomedical Sciences and Pathobiology, Virginia Maryland College of Veterinary Medicine, Virginia Tech, 245 Duck Pond Drive, Blacksburg, VA 24061, USA; klahmers@vt.edu

**Keywords:** mast cell tumor, dog, canine, proliferation indices, grade, prednisone, Ki67, AgNOR, mitotic count, mitotic index

## Abstract

Glucocorticoid administration is a common clinical practice that attempts to decrease the inflammation associated with and improve the resectability of canine mast cell tumors (MCTs). However, the impact of neoadjuvant glucocorticoids on the histological features and proliferation indices of canine MCTs is unknown. The objective of this study was to evaluate changes in tumor grade, mitotic count, Ki67, AgNOR, and AgNORxKi67 scores following short-course anti-inflammatory neoadjuvant prednisone in canine patients with MCTs. This was a prospective single-arm pilot study. Client-owned dogs with treatment-naïve cytologically confirmed MCTs were enrolled. Patients underwent an initial incisional biopsy followed by a 10–14-day course of anti-inflammatory prednisone and surgical resection. All histological samples were randomized, masked, and evaluated by a single pathologist. Unstained paired pre- and post-treatment samples were submitted to a commercial laboratory for Ki67 and AgNOR immunohistochemical analysis. There were 11 dogs enrolled with 11 tumors. There were no statistical differences between the pre- and post-treatment histological parameters of mitotic index, Ki67, AgNOR, or Ki67xAgNOR. There were no clinically significant alterations between pre-treatment and post-treatment in the assignment of tumor grades. A short course of anti-inflammatory prednisone does not appear to alter the histological parameters that affect grade determination or significantly alter the proliferation indices in canine MCTs.

## 1. Introduction

Mast cell tumors (MCTs) are the most common skin tumor affecting dogs, and the tumor grade is the most consistent and clinically relevant prognostic factor [1,2,3,4]. There are two MCT grading systems: the Patnaik system and the Kiupel system [3]. The Patnaik system categorizes tumors as grades 1, 2, or 3, with grade 1 as potentially curable and grade 3 conferring a poor prognosis [5]. The majority of MCTs, however, are classified as Patnaik grade 2, which demonstrate a wide range of biological behavior [3]. The Kiupel system was proposed to reclassify grade 2 tumors to improve clinical utility and categorizes all tumors as either low- or high-grade, with low-grade tumors having an excellent prognosis and high-grade tumors conferring a poor prognosis [6]. The recent consensus document by the Oncology-Pathology Working Group states that the two grading systems are complementary and recommends that the MCT grade is reported using both systems: G1/LG, G2/LG, G2/HG, or G3/HG [1].

Rapid cellular proliferation carries prognostic significance in malignancy and can be quantified by evaluating the mitotic count, broadly defined as the number of mitotic figures in 10 non-overlapping high-powered fields [7,8]. The mitotic count is a criterion in MCT grading, but the two grading systems have different cut-offs for grade determination, and interobserver variability and a lack of standardization of mitotic counts in canine MCTs have been reported [1,7,8,9]. Tumor proliferation involves not only cells in mitosis but also the number of cells actively engaged in the cell cycle and how quickly cells are progressing through the cell cycle. To that end, the mitotic count does not provide a global view of MCT proliferation, and additional markers evaluating the growth fraction and generation time can be employed to improve prognostication [10]. Ki67 is a nuclear protein that is expressed in all phases of the cell cycle except G0 and represents the tumor growth fraction. The relative number of cells expressing Ki67 is an established prognostic factor in canine MCTs and other malignancies in several species [2,11]. There are multiple variants of Ki67, with species, cell-type, and cell-cycle specificity, and it forms the perichromosomal layer during mitosis, preventing chromosomes from sticking together and maintaining chromosomal structural integrity. Following mitosis, Ki67 also functions in aiding in nucleolar organization [11]. Nucleoli are the sites of ribosome biogenesis and form around organizer regions (NORs), which contain tandem arrays of ribosomal gene repeats. The nucleolus is the largest non-membrane-bound subnuclear structure and can be easily visualized in the interphase nucleus [12,13]. A silver-based staining method is used to identify and quantify NORs, termed argyrophilic nucleolar organizing regions (AgNOR), and represents how quickly the cells progress through the cell cycle. Together, AgNOR and Ki67 further refine the anticipated biological behavior and can provide the clinician with information to guide treatment decisions for patients with tumors with the intermediate tumor grades of G2/LG and G2/HG [2].

Glucocorticoids are foundational in the treatment of canine MCTs. They are administered orally or intra-lesionally as a sole therapy or in combination with conventional chemotherapy, small-molecule-targeted therapy, radiation therapy, and/or surgery [14,15,16,17,18,19,20,21,22,23,24,25,26,27,28,29,30,31,32,33,34,35,36,37,38]. The majority of responses to glucocorticoids as a sole therapy are partial responses, with a response rate of approximately 70% [14,19,22,34,39,40,41]. Definitive local and local-regional therapy improve outcomes and quality of life, regardless of MCT location or grade [1,17,19,22,36,38,42,43,44,45,46,47,48,49,50,51,52,53,54,55]. It is a common clinical practice to attempt cytoreduction of MCTs with glucocorticoid therapy and to reduce the morbidity associated with definitive local therapy and/or provide a window of feasibility for curative-intent surgery.

The clinical response to glucocorticoids is attributed to their anti-inflammatory effects and the apoptosis of the mast cells via the activation of the glucocorticoid receptor (GCR) [15,56]. As a transcription factor, the activation of the cytosolic GCR (cGCR) results in changes in gene expression: anti-inflammatory and regulatory gene expression is transactivated, while pro-inflammatory gene expression is transrepressed [56]. Glucocorticoid administration also exerts its clinical effects through non-genomic mechanisms. The activation of cGCR results in the apoptosis of cells by targeting pro-survival factors for degradation [57]. Intracellular signaling is altered due to the release of proteins from the cGCR multi-protein complex upon the binding of glucocorticoids, resulting in rapid anti-inflammatory effects [56]. Glucocorticoids also act directly by negatively impacting cellular growth through the inhibition of arachidonic acid release or via direct interaction with cellular membranes. These direct interactions alter cellular physicochemical properties and the function of membrane-associated proteins, allowing the interference of cytokine synthesis, antigen processing, phagocytosis, and migration [56]. Glucocorticoid resistance in canine mast cells is reported to be related to an inhibition of GCR-mediated gene expression changes, increased cellular efflux, and an increase in anti-apoptotic factors [58]. The alteration of gene expression, cellular function, and intra- and intercellular communication by glucocorticoids has the potential to affect the number of cells actively in the cell cycle, how rapidly cells are progressing through the cell cycle, cellular and nuclear morphology, and the qualities of the tumor and stromal microenvironment.

The clinical implications of potential alterations in the MCT grade or the scoring of proliferation indices are significant. The prognosis and subsequent management of the patient after definitive local therapy is dichotomized by the MCT grade. Typically, low-grade tumors require no further treatment, even if incompletely excised, and patients are expected to have good to excellent outcomes, while patients with high-grade tumors are expected to succumb to their disease and require intensive multi-modal therapeutic strategies [1,2,6,35,36,37,38,39,48,52,53,54,59,60,61,62,63,64,65,66,67]. This then raises the question of how pre-operative glucocorticoid treatment may impact the histological parameters and criteria for grade determination and the immunohistochemical detection of proliferation indices in canine MCTs.

The objective of this study was to evaluate changes in tumor grade, mitotic count, Ki67, AgNOR, and AgNORxKi67 scores following short-course anti-inflammatory neoadjuvant prednisone in canine patients with MCTs. This pilot study was intended to guide hypothesis generation and future study design and assist in power analysis calculations regarding the impact of short-term neoadjuvant prednisone administration on the histological and proliferation indices in canine cutaneous mast cell tumors.

## 2. Materials and Methods

### 2.1. Study Population

Client-owned dogs presenting to the Virginia-Maryland College of Veterinary Medicine Veterinary Teaching Hospital (VMCVM) with naïve or recurrent cutaneous mast cell tumors were recruited. The inclusion criteria included a minimum body weight of 5 kg, a cytologic diagnosis of mast cell tumor by a board-certified clinical pathologist, a tumor size of ≥1 cm and <10 cm in the longest diameter, and an expected survival of ≥4 weeks without therapy. Prior surgery with mast cell tumor recurrence was allowed. The exclusion criteria included creatinine, ALT or AST ≥ 1.5x upper reference limit, albumin <2.0 g/dL, grade 2 or higher VCOG cytopenia, or concurrent or previous chemotherapy or kinase therapy, steroid administration, or radiation therapy. All clients were informed of the purpose of the study, and informed consent was obtained. This study was approved by the Virginia Tech Institutional Animal Care and Use Committee (IACUC) and the Veterinary Hospital Board.

### 2.2. Study Design

This was a prospective, single-arm, open-label pilot study. All procedures were performed at a single institution. All tumor measurements throughout the study were performed prior to manipulation, taken in three dimensions using digital calipers, and performed by the same investigator throughout the study (S.K.). The baseline evaluation included a physical exam, tumor measurements and photographs, CBC, a serum biochemistry panel, and urinalysis. Within seven days of the screening evaluation, a pre-treatment incisional 4–6 mm punch biopsy was performed under sedation using standard sedation protocols selected at the clinician’s discretion. The patients were discharged with oral prednisone at a targeted dose of 1.0 mg/kg administered once daily for 10–14 days [68]. The prescribed dose was adjusted to the nearest half-tablet size for owner convenience. The actual duration of treatment was adjusted based on clinic and surgeon availability. Concurrent treatment with H1- or H2-blocking agents was acceptable. Clients maintained and submitted a daily account of medication administration and observations. Prednisone was discontinued on the day of the excisional biopsy. An exam, tumor measurements and photographs, CBC, serum biochemistry, and urinalysis were performed prior to the excisional biopsy. Post-treatment tumor measurements were defined as the longest tumor diameter at the end of prednisone therapy but before excisional biopsy. Any noted adverse events were graded according to the VCOG-CTAE [69]. Gross surgical margins were recorded for each tumor. Curative-intent surgical margins were defined as either wide excision (lateral surgical margins >2 cm) or as lateral surgical margins proportional to the widest tumor diameter [54,70,71,72,73,74]. Surgical margins not meeting the definition of curative-intent were considered marginal excisions. Excisional biopsy and post-operative management were performed by or under the supervision of a Diplomate of the American College of Veterinary Surgeons (ACVS) per standard of care at the VMCVM.

### 2.3. Assessment of Histologic Parameters

Incisional pre-treatment and excisional post-treatment biopsy samples were processed in the standard preparation for routine histological evaluation. All samples were interpreted, and histologic margins reported by a Diplomate of the American College of Veterinary Pathologists (ACVP) for immediate clinical use. Upon the completion of all patient enrollment and participation, all samples were randomized, masked, and digitized by a non-investigator. The images were re-evaluated by a single board-certified (ACVP) pathologist (K.L.). All samples were graded according to the Patnaik and Kiupel grading systems and assigned to one of four possible categories: grade 1/low grade (G1/LG), grade 2/low grade (G2/LG), grade 2/high grade (G2/HG), and grade 3/high grade (G3/HG) [1]. The reported mitotic count was the number of mitoses per 10 high-powered fields (2.37 mm^2^) [7,8]. Complete histologic margins were defined in this study as ≥2 mm [72].

### 2.4. Assessment of Proliferation Indices

Unstained histological slides of paired incisional pre-treatment and excisional post-treatment biopsy samples were submitted to a commercially available reference lab (Michigan State University Diagnostic Center of Population and Animal Health (MSU DCPAH)) for immunohistochemical staining for Ki-67 and AgNOR. https://cvm.msu.edu/vdl/laboratory-sections/anatomic-surgical-pathology/biopsy-service/prognosis-of-canine-cutaneous-mast-cell-tumors (accessed on 29 May 2022). The results were reported per standard for all routine samples presented to MSU DCPAH.

### 2.5. Statistical Analyses

Continuous variables were analyzed with the paired t-test for normally distributed data or the Wilcoxon test for data that were not normally distributed. All *p*-values were 2-sided, and *p*-values <0.05 were considered statistically significant. Statistical analyses were performed with standard software (MedCalc Statistical Software version 18.1 (MedCalc Software bvba, Ostend, Belgium; 2018).

## 3. Results

### 3.1. Study Population and Tumor Details

Thirteen dogs were screened for enrollment. All dogs met the eligibility criteria, were enrolled, underwent incisional biopsy, and initiated prednisone treatment. Two dogs were removed from the study prior to excisional biopsy. One dog was removed from the study due to grade 4 gastrointestinal toxicity (gastrointestinal ulceration), and another dog was excluded due to a histological diagnosis on pre-treatment biopsy inconsistent with mast cell tumor. Eleven dogs completed the study with 11 paired tumor samples available for evaluation.

Patient and tumor details are listed in Table 1. Most of the tumors were novel (*n* = 9), and two dogs had recurrent mast cell tumors. The median age was 7.5 years (range, 3 years to 12 years). There were six castrated males and five spayed females. The median weight was 27.3 kg (4.3–45.3 kg). A variety of breeds were represented, with mixed-breed dogs being the most common (*n* = 4), and the remaining dogs each representing one breed. The majority of tumors were located on the trunk, tail, or limbs (*n* = 8), with one tumor each in the inguinal region, oral cavity, and ventral to the eye.

The median dose of prednisone was 0.8 mg/kg/day (range: 0.5–1.2 mg/kg). The median duration of prednisone administration was 11 days (range: 10–14 days). Pre-treatment, the median tumor volume was 2.89 cm^3^ (range: 0.8–160 cm^3^), and the median longest diameter (LD) was 21 mm (range: 14–92 mm). Post-treatment, the median tumor volume was 1.73 cm^3^ (range: 0.3–58.6 cm^3^), and the median LD was 17 mm (range: 10–60 mm). The overall response rate was 72.7%. Eight tumors decreased in size, one increased in size, and there was no change in size for two tumors. For the tumors that decreased in size, the median decrease in LD was 13.6 mm (range: 3–32 mm), with a median relative size decrease of 29% (17.6–47.4%).

### 3.2. Histological Parameters

Most of the tumors were amenable to curative-intent resection (*n* = 8), defined as wide surgical margins >2 cm (*n* = 6) or lateral surgical margins proportional to the widest tumor diameter (*n* = 2) [70,73]. Complete histologic margins were achieved in 54.5% (*n* = 6) of all tumors. All tumors with incomplete histologic margins were reported to have evidence of mast cells present at a surgical margin, i.e., no tumor margins classified as incomplete had “narrow” or clean histologic margins of <2 mm. Complete histologic margins were achieved in 62.5% (*n* = 5) of tumors resected with curative intent. All five of these tumors were resected with wide surgical margins >2 cm. One tumor increased in size following prednisone treatment (patient #6) and had incomplete histologic margins following curative-intent surgical resection with proportional margins. One tumor (patient #7) with complete histologic margins had been marginally resected. This tumor was located in the oral cavity, was G2/LG, and demonstrated the greatest reduction in tumor volume following prednisone treatment. The patient last presented to the VMCVM with non-MCT-related morbidity 47 months following resection, with no evidence of tumor recurrence.

Individual patient tumor grades and mitotic counts are listed in Appendix A. In pre-treatment tumor grade classification, G2/LG tumors were the most common (*n* = 9), and there was one tumor classified as G1/LG and one tumor classified as G2/HG. Post-treatment, 10 tumors were classified as G2/LG, and one tumor classified as G3/HG. Two tumors were interpreted to have a different grade following prednisone treatment (Figure 1). In both instances, the Patnaik designation increased, but the Kiupel designation did not change.

There were no appreciable inflammatory cells in any of the tumor samples, and differences between pre- and post-treatment could not be quantified. The median pre-treatment mitotic count was 2 per 10 hpf (range: 0–8), and the median post-treatment mitotic count remained 2 per 10 hpf (range: 0–25). There was no statistically significant difference between the mitotic counts pre- and post-treatment (*p* = 0.4210) (Figure 2). The median mitotic count excluding the G2/HG (pre-treatment) was 2 per 10 hpf (range: 0–3), and the median mitotic count excluding the G3/HG (post-treatment) was 1.5 per 10 hpf (range: 0–4). The one Kiupel high-grade tumor (patient #3) was noted to have an increase in the mitotic count post-treatment.

### 3.3. Proliferation Indices

All individual patient tumor proliferation indices pre- and post-treatment are listed in Appendix A. The median pre-treatment Ki67 score was 6.8 (1.8–32.4), and the median post-treatment score was 5.4 (1.6–35). Differences on an individual level varied, with most patients’ scores remaining roughly the same: increasing or decreasing within <3 points. Other patients’ scores markedly decreased (patients #4 and 10) or markedly increased (patient #7) following treatment. There was no statistically significant difference between the pre- and post-treatment cohort Ki67 scores (*p* = 0.4393) (Figure 3).

The median pre-treatment AgNOR score was 1.61 (1.1–3.07), and the median post-treatment score was 1.4 (0.93–2.8). Most individual paired tumors had similar AgNOR scores pre- and post-treatment, varying by less than 0.2 points. Three patients’ AgNOR scores decreased by >1 following treatment (patients #5, 8, and 10), although the difference in AgNOR scores between pre- and post-treatment cohorts did not reach significance (*p* = 0.0885) (Figure 4).

The median pre-treatment AgNORxKi67 product score was 14 (2.0–84.2), and the median post-treatment score was 7.2 (2.1–98). There was no statistically significant difference in the AgNORxKi67 score between treatment cohorts (*p* = 0.2046) (Figure 5). There was only one set of paired tumor samples with AgNORxKi67 scores above 54 (patient #3), whose tumor was classified as G2/HG pre-treatment and G3/HG post-treatment [10].

## 4. Discussion

The objective of this study was to evaluate changes in tumor grade, mitotic count, Ki67, AgNOR, and AgNORxKi67 scores following short-course anti-inflammatory neoadjuvant prednisone in canine patients with cutaneous mast cell tumors. The impetus for this study was the common clinical practice of neoadjuvant prednisone treatment in an attempt to cytoreduce mast cell tumors but always with the unanswerable question as to the impact on the tumor histopathology and the subsequent prognostication and treatment recommendations.

There is no consensus regarding the utility of pre-treatment biopsy in the initial screening evaluation of canine mast cell tumors [1]. The pre-treatment biopsy in our study had the potential to independently affect the histological parameters assessed following treatment with prednisone. Local inflammation and mast cell degranulation associated with a biopsy procedure could lead to alterations in the proliferation indices scores or in the criteria for tumor grading, such as the mitotic count, cellular or nuclear morphology, and edema or necrosis. In a recent study by Shaw et al., pre-treatment biopsy samples and subsequent excisional biopsy samples had a very high level of concordance using the Patnaik grading system and a high level of concordance using the Kiupel grading system [75]. This was true regardless of the tumor location or the biopsy technique employed: wedge, punch, or needle; specifically, the punch biopsy had 100% agreement under the Patnaik system and 95% agreement under the Kiupel system. The mean duration between the pre-treatment and excisional biopsies in that study was 14 days, and the median was 9 days (2–111 days). All pre-treatment biopsies in our study were performed using a 4 mm or 6 mm punch instrument with a median duration between the pre-treatment biopsy and excisional biopsy of 11 days (10–14 days). This suggests that the pre-treatment biopsy procedure in our study likely had minimal impact on the histologic parameters of the excisional biopsy.

The tumor grade and mitotic count are the most consistent prognostic factors in canine mast cell tumors [1,9,39,76,77,78,79]. Clinically, the prognosis and treatment recommendations can be dichotomized based on tumor grade: G1/LG and G2/LG tumors confer an excellent prognosis, typically with no additional therapy required, while G2/HG and G3/HG tumors consistently result in a 1-year survival rate of <50%, even with additional local or systemic therapy [1,80]. In our study, there was no statistically significant difference in the median mitotic count or in the tumor grade classification following treatment with prednisone. The tumor grade classification was altered following treatment for two tumors. In both cases, the Patnaik assignment increased but the Kiupel assignment did not change, and the overall change in tumor grade classification had no clinical impact. Our findings are consistent with those of a recently published study in which there was no statistically significant difference in the mitotic count following prednisone treatment, and 2/13 paired tumor samples following prednisone treatment had altered Patnaik grades without changes in the Kiupel grades [40]. All but one of the tumors in our study was classified as low-grade. Additional studies restricted to the impact of pre-treatment with prednisone on high-grade tumor mitotic count and grade classification are indicated. It is important to note that tumor grade classification using the Patnaik system is subject to inter-observer variability, which is reported to have 50–60% discordance, while the Kiupel system reports 96–98% consistency among pathologists [1,6,81]. Furthermore, the determination of the mitotic count can be subject to individual variation [7,8]. In our study, a single pathologist interpreted all tumor samples after they were digitized, randomized, and masked by a non-investigator, thus controlling for both inter-observer variability and bias due to knowledge of the sample origin or treatment status.

The proliferation indices, AgNOR and Ki67, and their product, AgNORxKi67, have demonstrated utility in refining the prognosis of canine mast cell tumors, especially in intermediate-grade tumors [2,10,39,60,79,82,83,84,85]. Clinically, these indices may also be useful in determining whether adjuvant therapy is warranted following surgical excision, as increasing AgNOR and Ki67 scores have been associated with an increased risk of local tumor recurrence and metastasis [10,39,60,83,85]. Most canine mast cell tumors are intermediate-grade tumors (Patnaik grade 2), which are now also classified as either Kiupel grade high or Kiupel grade low [1]. Clinically, there is no standard of care for intermediate-grade tumors, and the proliferation indices provide complementary information that is used to guide ancillary treatment decisions. Therefore, understanding how the administration of routine peri-operative medications impacts these scores is important during the initial treatment planning. A recent study evaluated the impact of opioid administration on histologic parameters, including the proliferation indices, in canine cutaneous mast cell tumors, but the impact of prednisone has not been previously evaluated [40,86]. In our study, there was no trend noted at the individual level, and no statistically significant difference was noted in the paired tumor samples for the Ki67 score. On an individual level, most patients’ AgNOR scores varied by <0.2 points between paired samples. However, there were three patients whose AgNOR scores decreased by >1 following prednisone treatment. This did not reach statistical significance (*p* = 0.08), which could be due to type II error. A post hoc power analysis was performed, and 27 paired samples would be required to detect a mean difference of 0.314 in the AgNOR score with α = 0.05 and β = 20. There are few studies that evaluate the AgNOR score as an independent prognostic factor, and unfortunately, AgNOR was not an immunohistochemical marker that was included in a recent systematic review and meta-analysis [2]. All existing studies have demonstrated via multi-variate analyses that an increasing AgNOR score is associated with an increased risk of local tumor recurrence, distant tumor occurrence, lymph node metastasis, and/or MCT-related mortality [10,82,85]. Based on our findings, larger randomized controlled studies evaluating the impact of prednisone on the AgNOR score and long-term follow-up would be warranted. It is important to note that in the available literature there is variability in the methodology and cut-off points for assessing the proliferation indices, particularly Ki67 [2]. We evaluated AgNOR and Ki67 in this study via sample submission to an external commercial laboratory. This provided consistent, unbiased, validated, and reproducible data that have practical and applicable relevance.

Glucocorticoids are used as a sole therapy in cutaneous mast cell tumors, administered orally or intra-lesionally. Existing studies classify most responses as partial responses, with all but one study reporting overall response rates between 63 and 75% [14,19,22,39,40,41]. All responses in our study were partial responses, with an overall response rate of 72.7%. For the tumors that responded, the median decrease in tumor LD was 1.36 cm, and the relative decease in tumor volume was 29%. The response to glucocorticoid administration has been associated with larger tumor size and low-grade classification [22,39]. Four of the tumors in our study were >5 cm^3^, and nearly all of them were Kiupel low-grade. It is possible that the MCT response to glucocorticoids is due to the anti-inflammatory effects rather than mast cell apoptosis; there was no appreciable inflammation in any of the tumor samples, and no differences were noted between the pre- and post-treatment samples.

It is consistently reported that complete surgical resection is the treatment of choice for mast cell tumors, and complete histologic margins may be considered curative for low-grade tumors [1,22,38,41,42,43,44,47,50,51,53,54,59,60,61,64,65,70,71,72,73,74,83,87,88,89,90,91,92,93,94,95,96,97,98]. Mast cell tumors can be deemed non-resectable or not amenable to curative-intent surgical margins. In a recent study, there was a significant association with increased risk of post-operative complication in patients with MCTs and incomplete histologic margins [99]. It is reasonable to attempt cytoreduction with pre-operative glucocorticoid administration in patients with non-resectable or marginally resectable MCTs. The concerns regarding glucocorticoid treatment for cytoreduction include whether surgical margins based on post-treatment tumor size would yield complete histologic margins and whether treatment would be associated with an increased risk of post-operative incisional complications. In our study and others, surgical margins based on tumor size following pre-operative treatment with glucocorticoids have yielded complete histologic margins [22,40,99]. A long-term prospective evaluation of these patients is warranted to determine whether the local tumor recurrence rate is impacted [54]. Post-operative complications following MCT resection, whether wide or intentionally marginal, are reported to be 13–29% [41,99]. Although not evaluated in our study, others have reported that dogs treated with pre-operative glucocorticoids have not had an increased risk of post-operative complications [22,40,99].

There are several limitations to our study. This study was intended to generate data to guide hypothesis generation and future study design and assist in power analysis calculations regarding the impact of short-term neoadjuvant prednisone administration on the histological and proliferation indices in canine cutaneous mast cell tumors. As such, there was no control group, the patients did not undergo standardized staging evaluations, and there are no long-term outcome data. The identification of tumor grade was not a study enrollment criterion, resulting in a paucity of high-grade tumors in our study participants. As such, the findings in this study are applicable to low-grade tumors and should not be extrapolated to high-grade tumors. Glucocorticoid resistance mechanisms in canine mast cells are related to the inhibition of GCR-mediated gene expression changes and an increase in anti-apoptotic factors [58]. It is possible that as a cohort the tumors in our study failed to demonstrate significant changes due to variation in the gross tumor response to glucocorticoid therapy. Future studies may elect to stratify tumor cohorts or restrict inclusion criteria based on the clinical response to glucocorticoids to maximize the identification of tumor histological or proliferation alterations following treatment. Our pilot study focused on commercially accessible parameters of routine tumor histopathology and the immunohistochemical detection of AgNOR and Ki67. However, a multi-faceted methodological approach evaluating the impact on gross and histologic tumor parameters, differential gene expression, and protein expression may be warranted in future studies to better characterize the role and impact of glucocorticoids in MCT management. Additionally, the evaluations of changes in the tumor volumes, the histologic margins, and the relationship between the surgical dose and histologic margins were secondary and exploratory objectives of our study and should be interpreted with caution. Tumor margins were not re-evaluated by a single investigator-pathologist, which introduces the potential for inter-observer error, and the tumor volumes post-treatment must be interpreted carefully in light of the pre-treatment biopsy, in that the decrease in tumor volume and LD may be impacted by the pre-treatment biopsy two weeks prior to the final measurement [100]. While our findings corroborate that a pre-treatment biopsy does not seem to impact the tumor grade or mitotic count, the pre-treatment biopsy procedure could have independently affected the proliferation indices, which were not evaluated in a recent study, and larger prospective randomized placebo-controlled studies are necessary for further investigation [40,75].

## 5. Conclusions

The data from this study provide the catalysts and foundation for the next steps into the investigation of the role glucocorticoids in canine MCT management. The results indicate that there appear to be no clinically relevant alterations in the tumor grade classification, mitotic count, or the proliferation indices in low-grade mast cell tumors, three criteria consistently relied-upon in the management of canine cutaneous mast cell tumors. Decreases in the proliferation index of AgNOR warrant further investigation, and randomized placebo-controlled appropriately powered studies are necessary to confirm our results. The findings of our study can guide patient and tumor selection criteria for future studies, including stratifying cohorts based on tumor grade, size, and the response to glucocorticoid therapy. This study may also provide an impetus for a multi-tiered molecular investigation to characterize the global impact of neoadjuvant glucocorticoid therapy in canine MCTs. Finally, this study provides support for a tangentially related investigation regarding factors associated with the response to prednisone therapy and the long-term impact of neoadjuvant prednisone administration on the surgical dose and the resulting histologic margins.

## Figures and Tables

**Figure 1 vetsci-09-00277-f001:**
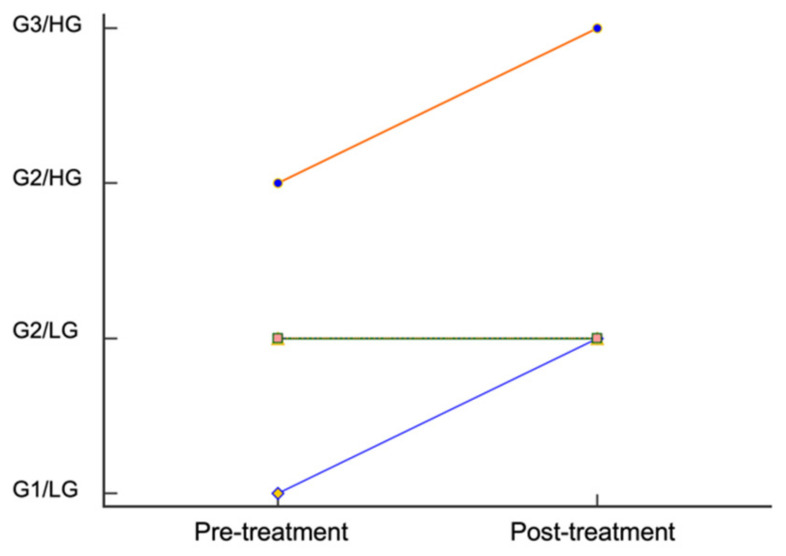
Pre- and post-treatment tumor grade classifications. The classification for nine G2/LG tumors did not change. The Patnaik classification, but not the Kiupel designation, increased for two tumors following treatment with prednisone.

**Figure 2 vetsci-09-00277-f002:**
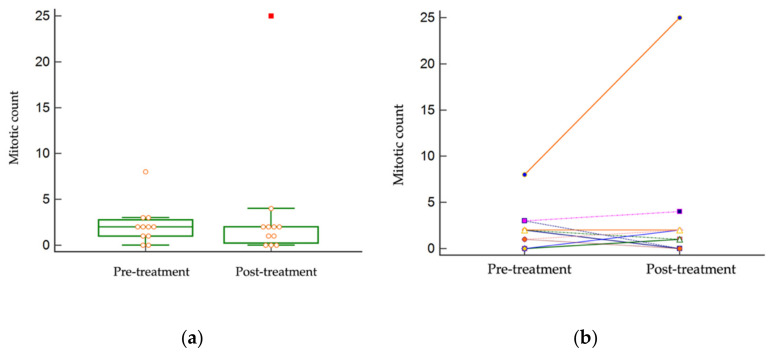
Mitotic counts in paired tumor samples. There was no statistically significant difference between pre- and post-treatment mitotic counts (*p* = 0.4210) (**a**) Distribution of mitotic counts pre- and post-treatment. (**b**) Individual tumor paired mitotic counts.

**Figure 3 vetsci-09-00277-f003:**
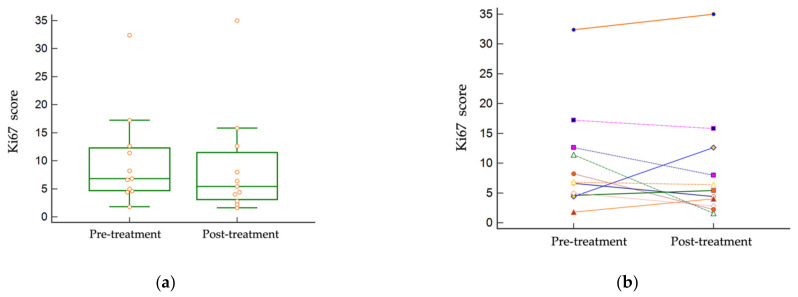
Ki67 scores in paired tumor samples. There was no statistically significant difference between pre- and post-treatment Ki67 scores (*p* = 0.4393). (**a**) Distribution of Ki67 scores pre- and post-treatment. (**b**) Individual tumor paired Ki67 scores.

**Figure 4 vetsci-09-00277-f004:**
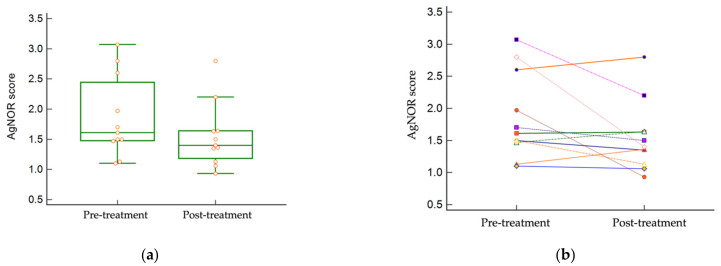
AgNOR scores in paired tumor samples. There was no statistically significant difference between pre- and post-treatment AgNOR scores (*p* = 0.0885). (**a**) The median AgNOR score pre- and post-treatment. (**b**) Individual tumor paired AgNOR scores. Three patients had a decrease >1 in the AgNOR score following treatment.

**Figure 5 vetsci-09-00277-f005:**
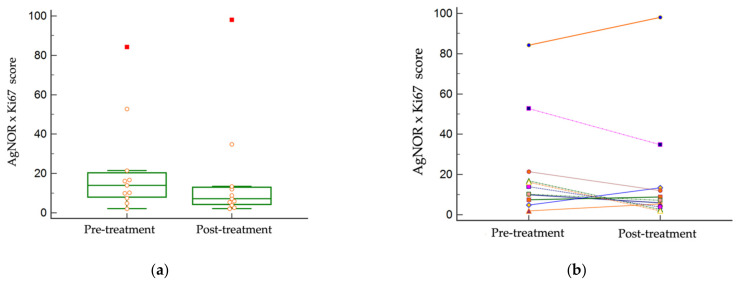
AgNORxKi67 scores in paired tumor samples. There was no statistically significant difference between pre- and post-treatment AgNORxKi67 scores (*p* = 0.2046). (**a**) The distribution of AgNORxKi67 scores pre- and post-treatment. (**b**) Individual tumor paired AgNORxKi67 scores.

**Table 1 vetsci-09-00277-t001:** Patient demographics and tumor details.

Patient #	Breed	Age (y)	Sex	Weight (kg)	Recurrent or Novel	Tumor Location	Tumor Volume (mm^3^)	Tumor Volume (% Change)	Tumor Grade	Surgical Margins ^1^ and Histologic Margins ^2^
1	Golden retriever	7.5	FS	34.5	Novel	Proximal lateral left forelimb	pre: 2205	−49	Pre: G2/LG	Wide
							Post: 1125		Post: G2/LG	Complete
2	Mixed	3.2	FS	45.3	Novel	Tail	Pre: 1425	−64.9	Pre: G2/LG	Wide
							Post: 500		Post: G2/LG	Complete
3	Yorkshire terrier	5.3	MC	4.3	Novel	Ventral to left eye	Pre: 1078	0	Pre: G2/HG	Marginal
							Post: 1078		Post: G3/HG	Incomplete
4	Staffordshire terrier	6.7	MC	27.3	Novel	Interdigital	Pre: 3300	−22.7	Pre: G2/LG	Marginal
							Post: 2550		Post: G2/LG	Incomplete
5	Mixed	11.6	MC	30.6	Recurrent	Left abdomen	Pre: 160,080	−75.4	Pre: G2/LG	Wide
							Post: 39,360		Post: G2/LG	Complete
6	Miniature Schnauzer	9.6	MC	10.1	Novel	Right dorsal tarsus	Pre: 588	200.7	Pre: G2/LG	Proportional
							Post: 1768		Post: G2/LG	Incomplete
7	Mixed	8.3	MC	12.4	Novel	Distal medial left hindlimb	Pre: 765	−60.8	Pre: G1/LG	Proportional
							Post: 300		Post: G2/LG	Incomplete
8	Norwegian elkhound	9.3	FS	24.9	Novel	Oral cavity	Pre: 9996	−82.7	Pre: G2/LG	Marginal
							Post: 1729		Post: G2/LG	Complete
9	German shorthair pointer	3.5	MC	27.4	Novel	Proximal lateral right hindlimb	Pre: 2890	−56.4	Pre: G2/LG	Wide
							Post: 1260		Post: G2/LG	Incomplete
10	Mixed	3.9	MC	38.9	Recurrent	Distal lateral right hindlimb	Pre: 58,608	0	Pre: G2/LG	Wide
							Post: 58,608		Post: G2/LG	Complete
11	Beagle	8.2	FS	19.8	Novel	Left inguinal region	Pre: 9620	−57.3	Pre: G2/LG	Wide
							Post: 4104		Post: G2/LG	Complete

^1^ Wide excision is defined as ≥2 cm surgical margins, and proportional excision is defined as lateral surgical margins proportional to the widest tumor diameter. ^2^ Complete excision is defined as ≥2 mm histologic margins.

## Data Availability

All data generated or analyzed during this study are included in this article. The datasets used for analysis are available from the corresponding author upon request.

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
