# Peer review of "Evaluation of Tumor Grade and Proliferation Indices before and after Short-Course Anti-Inflammatory Prednisone Therapy in Canine Cutaneous Mast Cell Tumors: A Pilot Study"

_vetsci, 2022, doi:10.3390/vetsci9060277_

Round 1

Reviewer 1 Report

The manuscript is well organised, even if of low novelty. I would suggest to add":A Pilot Study" to the title, because it is based on a limited number of cases and, as the authors state, there are several limitationa to their study, which lacks follow up data, controls and long-term staging. The results obtained are applicable to low-stage tumors but cannot be applied to high stage tumors, because only 2 of the latter were present in the study.

In my opinion the number of cases should be increased, in order to have more precise and useful data, and not only presumptive.

Author Response

Dear Editor and Reviewers,

Thank you for the opportunity to respond to the comments and suggestions on the manuscript entitled “Evaluation of tumor grade and proliferation indices before and after short-course anti-inflammatory prednisone therapy in canine cutaneous mast cell tumors”.  The reviewer comments were thoughtful and respectful, and we believe addressing them clarifies the study and makes the article stronger.

Response to reviewer 1:

The number of dogs enrolled and tumors evaluated is commonly noted by all reviewers.  Reviewers #1 and #2 suggested adding “ :A Pilot Study” to the article title.  We agree and have modified the manuscript title.

Reviewer 2 Report

This manuscript reports a pilot study about the influence of anti-inflammatory therapy in tumour grade and proliferation indices. It is interesting, but with a very low number of cases. The results probably were affected by this issue. To me no valid conclusions can be made but this article can be the beggining of a more consisted study for the authors. Moreover  this study is well written and designed. The title of the article must include “pilot study”.  

Author Response

Dear Editor and Reviewers,

Thank you for the opportunity to respond to the comments and suggestions on the manuscript entitled “Evaluation of tumor grade and proliferation indices before and after short-course anti-inflammatory prednisone therapy in canine cutaneous mast cell tumors”.  The reviewer comments were thoughtful and respectful, and we believe addressing them clarifies the study and makes the article stronger.

Response to reviewer 2:

The number of dogs enrolled and tumors evaluated is commonly noted by all reviewers.  Reviewers #1 and #2 suggested adding “ :A Pilot Study” to the article title.  We agree and have modified the manuscript title.

Reviewer 3 Report

The manuscript "Evaluation of tumor grade and proliferation indices before and after short-course anti-inflammatory prednisone therapy in canine cutaneous mast cell tumors" is interesting and addresses a frequent problem related to the need to reduce tumor volume before performing surgery. In this regard, I have some doubts and observations that need to be answered.

Introduction

-the introduction is clear and adequately leads to the objective of the study

Methodology

-the sample size is small, however, the authors adequately discuss the weaknesses of this

-explain why prednisone therapy was temporally variable (between 10 and 14 days)

-I suggest to evaluate and quantify inflammatory cells in HE sections

-describe in detail the immunohistochemical methodology for the detection of ki67 and the histochemical technique for AgNORs. Describe also the method of quantification of both parameters

Results

-explain why different doses of prednisone was used (0,5-1,2 mg/kg). Will this range induce variability in the response?

-include a column of histological grade by patient in Table I

-I suggest to show some representative microphotographs of ki67 expression and AgNORs pre and post therapy

-legends of Figure 3-5 should include p-value obtained

Discussion

-Based on the fact that treatment with prednisone induced a 72.7% overall response, but no changes were observed in the proliferative indices, what could be the reason for the observed clinical response? anti-inflammatory effect? Please, discuss this point

Author Response

Dear Editor and Reviewers,

Thank you for the opportunity to respond to the comments and suggestions on the manuscript entitled “Evaluation of tumor grade and proliferation indices before and after short-course anti-inflammatory prednisone therapy in canine cutaneous mast cell tumors”.  The reviewer comments were thoughtful and respectful, and we believe addressing them clarifies the study and makes the article stronger.

The number of dogs enrolled and tumors evaluated is commonly noted by all reviewers. Reviewers #1 and #2 suggested adding “ :A Pilot Study” to the article title.  We agree and have modified the manuscript title. 

  • Explain why prednisone therapy was temporally variable (between 10 and 14 days).
  • Explain why different doses of prednisone was used (0,5-1,2 mg/kg). Will this range induce variability in the response?
    • The explanation of dosing and duration of treatment should be clarified for the reader and to improve reproducibility of the study. 
    • The target dose for all dogs was within the anti-inflammatory range: 0,5-1,0 mg/kg/day. The prescribed dose was adjusted for owner convenience to the nearest half-tablet.  Prednisone is available in 5mg or 20mg tablets.  The immune-suppressive range of prednisone dosing is 2,0mg/kg/day.  We do not anticipate that the dose range within the study population would induce variability of response, as all dosing falls within the “anti-inflammatory” range commonly used clinically for prednisone.
    • Similarly, the duration of treatment was adjusted for clinical business days: specifically, due to weekend/holidays as well as surgeon scheduling. The target range was 10-14 days, the time frame reported previously in the literature, and the timeframe reported to correspond with median maximal MCT response to prednisone therapy: the response rates of 50% and greater are noted with shorter treatment durations, the median time to maximal tumor regression is reported to be 14-days[1, 2].
    • The reference for determining treatment duration has been cited in the Methods (line 138). Statements clarifying target and prescribed dose, as well as the actual duration of treatment have been added to the Methods, as well (lines 138-140).
  • I suggest to evaluate and quantify inflammatory cells in HE sections
    • The authors appreciate this suggestion and understand the rationale to evaluate inflammatory cells in the pre- and post-treatment samples. There was no appreciable inflammation in any of the samples, and statements have been added to the manuscript (lines 236-237, and lines 367-370).
  • describe in detail the immunohistochemical methodology for the detection of ki67 and the histochemical technique for AgNORs. Describe also the method of quantification of both parameters
  • I suggest to show some representative microphotographs of ki67 expression and AgNORs pre and post therapy
    • The IHC for Ki67 and AgNOR are commercial tests and the results for this study were not a collaborative effort. Detailed methodology nor representative microphotographs for the IHC performed as contractual service in our study through the Michigan State University Veterinary Diagnostic Laboratory are available for publication.  I have added a link to the commercial laboratory sample submission site in the Methods (lines 172-173).
  • include a column of histological grade by patient in Table I
    • Agreed, this improves the article’s readability. A column has been added as suggested.
  • legends of Figure 3-5 should include p-value obtained
    • Thank you; the p-values have been added as suggested, as well as to the legend of Figure 2.
  • Based on the fact that treatment with prednisone induced a 72.7% overall response, but no changes were observed in the proliferative indices, what could be the reason for the observed clinical response? anti-inflammatory effect? Please, discuss this point
    • It is indeed interesting that prednisone is known to induce a clinical response in canine MCT, and it was the intent of our study to lay the foundation for elucidating quantifiable alterations in clinically-relevant indices. It is possible that the observed clinical response is related to changes in the proliferation indices, specifically AgNOR.  The study population is small, n=11 paired samples, and type II error may be partially responsible for the findings of our study.  To detect a statistically-significant difference following prednisone treatment, a sample of 27 paired samples would be necessary (discussed on lines 346-348).  The MCT response to prednisone is attributable to anti-inflammatory effects and apoptosis of the mast cells via activation of the glucocorticoid receptor (GCR).  Apoptosis of mast cells would not necessarily be expected to alter the proliferation indices, as activation of the GCR results in apoptosis via targeting pro-survival factors for degradation.  Similarly, an equally-interesting question is the underlying resistance to prednisone therapy in some MCT.  One study evaluating resistance mechanisms in mast cells suggests that it is related to inhibition of GCR-mediated gene expression changes and by increasing anti-apoptotic factors,  It is possible that, in future studies, population stratification to evaluate only tumors that respond to prednisone may be indicated to better evaluate proliferation indices.  In general, however, the MCT response to prednisone is multifactorial, with genomic and non-genomic mechanisms; this is discussed in detail in the introduction on lines 80-99.  In vitro studies would better address the mechanistic underpinnings of prednisone on the proliferation indices, and our study would help guide study design as well as hypothesis generation (lines 406-409).

Reviewer 4 Report

Comments on the manuscript VetSci 1734668 entitled “Evaluation of tumor grade and proliferation indices before and after short-course anti-inflammatory prednisone therapy in ca- 3 nine cutaneous mast cell tumors”, by Klahn et al.

This paper describes a study on canine mast cell tumours. This is an interesting article, and the main criticism is the low number of animals enrolled. There are some limitations on the study that authors clearly describe in the manuscript.

Minor Revisions

  1. Introduction, line 47: The “2.37mm2” is an area of the objective of a specific microscope. But we can evaluate the mitotic count with other micrsocopes with different areas. So, the “2.37mm2” should be removed from the manuscript.

  1. Material and methods, lines 162-163: Please insert in the manuscript the area of the objective used to evaluate the mitotic count (and also for ki-67 and AgNors).

  1. References: All but 3 references (references 32, 57, and 64) have the name of the journals in full; please put all the journals in full or abbreviated.

Author Response

Dear Editor and Reviewers,

Thank you for the opportunity to respond to the comments and suggestions on the manuscript entitled “Evaluation of tumor grade and proliferation indices before and after short-course anti-inflammatory prednisone therapy in canine cutaneous mast cell tumors”.  The reviewer comments were thoughtful and respectful, and we believe addressing them clarifies the study and makes the article stronger.

The number of dogs enrolled and tumors evaluated is commonly noted by all reviewers.  Reviewers #1 and #2 suggested adding “ :A Pilot Study” to the article title.  We agree and have modified the manuscript title.

  • Introduction, line 47: The “2.37mm2” is an area of the objective of a specific microscope. But we can evaluate the mitotic count with other micrsocopes with different areas. So, the “2.37mm2” should be removed from the manuscript.
    • Thank you for your suggestion. We agree that in the introduction, the definition of mitotic count can be more broad, and we have slightly modified the statement in the introduction (lines 47-48).  We did, however, specify in the Methods the area of the objective used to evaluate mitotic count in our study (lines 164-165), in line with a recent article on reporting guidelines for manuscripts regarding prognostic markers[3]. 
  • Material and methods, lines 162-163: Please insert in the manuscript the area of the objective used to evaluate the mitotic count (and also for ki-67 and AgNors).
    • We have added this detail in the Methods (lines 164-165). However, the Ki-67 and AgNOR IHC was performed by a commercial lab, and not through a collaboration.  Details on IHC methodology, including evaluation, are unavailable.  I have added a link to the commercial lab submission website for additional clarification (lines 172-173).
  • References: All but 3 references (references 32, 57, and 64) have the name of the journals in full; please put all the journals in full or abbreviated.
    • References 32, 57, and 64 have been updated to list the name of the journals in full.

[1]        J. Dobson, S. Cohen, and S. Gould, "Treatment of canine mast cell tumours with prednisolone and radiotherapy," (in eng), Veterinary and comparative oncology, vol. 2, no. 3, pp. 132-41, Sep 2004, doi: 10.1111/j.1476-5810.2004.00048.x.

[2]        S. P. Teng, W. L. Hsu, C. Y. Chiu, M. L. Wong, and S. C. Chang, "Overexpression of P-glycoprotein, STAT3, phospho-STAT3 and KIT in spontaneous canine cutaneous mast cell tumours before and after prednisolone treatment," (in eng), Veterinary journal (London, England : 1997), vol. 193, no. 2, pp. 551-6, Aug 2012, doi: 10.1016/j.tvjl.2012.01.033.

[3]        F. Y. Schulman et al., "Reporting guidelines for manuscripts on tumor prognosis," Veterinary pathology, vol. 59, no. 3, pp. 397-398, 2022, doi: 10.1177/03009858221082207.

Round 2

Reviewer 1 Report

The manuscript is well structured. It is a real pity that the number of cases is low. In my opinion, it has affected the statistical results. It would be better to collect more cases to emeliorate the results.

Reviewer 2 Report

The report as pilot study is apropriate.